# SPATIO-TEMPORAL SELF-ATTENTION FOR EGOCENTRIC 3D POSE ESTIMATION

## ABSTRACT

Vision-based ego-centric 3D human pose estimation (ego-HPE) is essential to support critical applications of $x$R-technologies. However, severe self-occlusions and strong distortion introduced by the fish-eye view from the head mounted camera, make ego-HPE extremely challenging. While current state-of-the-art (SOTA) methods try to address the distortion, they still suffer from large errors in the most critical joints (such as hands) due to self-occlusions. To this end, we propose a spatio-temporal transformer model that can attend to semantically rich feature maps obtained from popular convolutional backbones. Leveraging the complex spatio-temporal information encoded in ego-centric videos, we design a spatial concept called *feature map tokens* (FMT) which can attend to all the other spatial units in our spatio-temporal feature maps. Powered by this FMT-based transformer, we build Egocentric Spatio-Temporal Self-Attention Network (Ego-STAN), which uses heatmap-based representations and spatio-temporal attention specialized to address distortions and self-occlusions in ego-HPE. Our quantitative evaluation on the contemporary sequential $x$R-EgoPose dataset, achieves a 38.2% improvement on the highest error joints against the SOTA ego-HPE model, while accomplishing a 22% decrease in the number of parameters. Finally, we also demonstrate the generalization capabilities of our model to real-world HPE tasks beyond ego-views.

## 1 INTRODUCTION

The rise of virtual immersive technologies, such as augmented, virtual, and mixed reality environments ($x$R) [1–3], has fueled the need for accurate human pose estimation (HPE) to support critical applications in medical simulation training [4] and robotics [5], among others [6–10]. Vision-based HPE has increasingly become a primary choice [11–15] since the alternative requires the use of sophisticated motion capture systems with sensors to track major human joints [16], impractical for real-world use. Vision-based 3D pose estimation is largely divided on the basis of camera viewpoint: outside-in versus egocentric view. Extensive literature is devoted to outside-in 3D HPE, where the cameras have a fixed effective recording volume and view angle [17–20], which are unsuitable for critical applications where higher and robust (low variance) accuracies are required [4]. In contrast, the egocentric perspective is mobile and amenable to large-scale cluttered environments since the viewing angle is consistently on the subject with minimal obstructions from the surroundings [21–23].

Nevertheless, the egocentric imaging does come with challenges: lower body joints are (a) visually much smaller than the upper body joints (*distortion*) and (b) in most cases heavily occluded by the upper torso (*self-occlusion*). Recent works address these challenges by utilizing the dual-branch autoencoder-based 2D to 3D pose estimator [21], and by incorporating extra camera information [24]. However, self-occlusions remain challenging to address from only static views. Moreover, while critical applications of ego-HPE (surgeon training [4]) require accurate and robust estimation of extremities (hands and feet), the current methods suffer from high errors on these very joints, making them unsuitable for these critical applications [21, 24]. Previous outside-in spatio-temporal works attempt to regress 3D pose from an input sequence of 2D keypoints – not images – [25–28], and focus on mitigating the high output variance of 3D human pose. In contrast, we estimate accurate 2D heatmaps – from images – by dynamically aggregating on intermediate feature maps and consequently produce accurate 3D pose. Moreover, outside-in occlusion robustness methods are not applicable in ego-pose due to dynamic camera angles, constantly changing background, and distortion, with

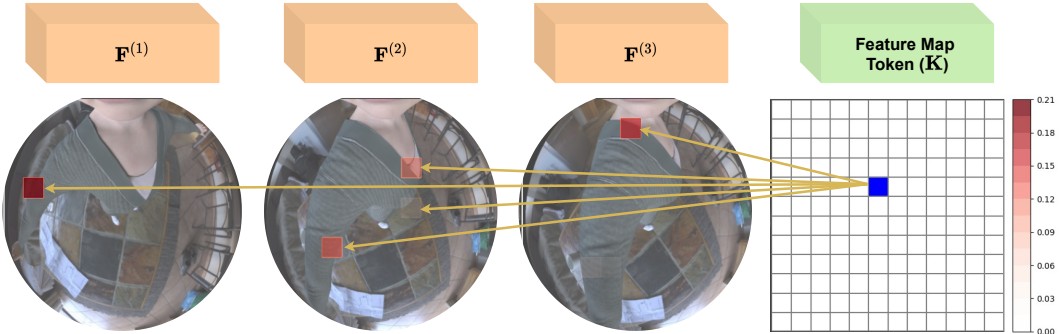

Figure 1: **Interpreting Ego-STAN's Attention Mechanism.** A sequence of images $\mathbf{I}^{(1)}$, $\mathbf{I}^{(2)}$, and $\mathbf{I}^{(3)}$, yields feature maps $\mathbf{F}^{(1)}$, $\mathbf{F}^{(2)}$, and $\mathbf{F}^{(3)}$, respectively, and are appended with a (learnable) feature map token ($\mathbf{K}$). Sections of Ego-STAN's feature map tokens (in blue) can be deconvolved to identify the corresponding attended region(s) in the image sequence (in red), to allow the interpretation of information aggregation from the images.

recent works requiring supervision on joint visibility [29–31]. Therefore, we need to build specialized models to address the unique challenges of ego-HPE.

Given these challenges, we investigate the following question: *how can we design a unified model to reliably estimate the location of heavily occluded joints and address the distortions in ego-centric views*? To this end, we propose **Egocentric Spatio-Temporal Self-Attention Network (Ego-STAN)** which leverages a specialized spatio-temporal attention, which we call *feature map token* (FMT), heatmap-based representations, and a simple 2D to 3D pose estimation module. On the SOTA sequential ego-views dataset $x$R-EgoPose [21], it achieves an average **improvement of 38.2%** mean per-joint position error (MPJPE) on the highest error joints against the SOTA egocentric pose estimation work [21] while **reducing 22% trainable parameters** on the $x$R-EgoPose dataset [21]. Furthermore, Ego-STAN genralizes to other HPE tasks in static ego-views Mo$^2$Cap$^2$ dataset [22], and outside-in views on the Human3.6M dataset [16] where it reduces the MPJPE by 9% against [21], demonstrating its ability to generalize to real-world views and adapt to other HPE scenarios. Our main contributions are summarized as follows.

- **Feature map token and interpreting attention.** To leverage the complex spatio-temporal information encoded in ego-centric videos, we design *feature map token* (FMT), learnable parameters that, alongside our spatio-temporal Transformer, can globally attend to all spatial units of the extracted sequential feature maps to draw valuable information. FMT also provides interpretability, revealing the complex temporal dependence of the attention (Fig. 1).

- **Hybrid Spatio-temporal Transformer powered by FMT.** Powered by the FMT, we design **Ego-STAN**'s hybrid architecture which utilizes spatio-temporal attention endowed by the FMT and Transformers [32] to self-attend to a sequence of semantically rich feature maps extracted by Convolutional Neural Networks (ResNet-101) [33]. Complementary to this architecture, we also propose an $\ell_1$-based loss function to accomplish robust pose estimation, handling both self-occlusions and visibly difficult (low resolution) joints. In addition, we also evaluate Ego-STAN on the Human3.6M, an outside-in sequential HPE dataset, showing an improvement of 8% on Percentage of Correct Keypoint (PCK) of 2D joint detection demonstrating the versatility of the proposed attention architecture and FMT.

- **Direct regression from heatmap to 3D pose.** We propose a simple neural network-based 2D heatmap to 3D pose regression module, which significantly reduces the overall MPJPE and the number of trainable parameters as compared the SOTA [21]. We also indirectly evaluate the advantages of this module via HPE on the Mo$^2$Cap$^2$ dataset (static ego-HPE) and on the Human3.6M dataset. Using detailed ablations, we also reveal a surprising fact: the auto-encoder-based architectures recommended by SOTA may be creating information bottlenecks and be counterproductive for ego-HPE.

- **Extensive ablation studies.** We perform comprehensive ablation studies to analyze the impact of each component of Ego-STAN. These ablations thoroughly demonstrate that the composition of the Transformer network, $\ell_1$ loss, Direct 3D regression, and the FMT, lead to the superior performance of Ego-STAN.

## 2 RELATED WORK

This section discusses related work on 3D HPE, for both static (single frame) and sequential (multi-frame) models, alongside Transformer-based self-attention, on two specific camera viewpoints: (1) an outside-in viewpoint, the image capture of a subject from a distance, (2) an egocentric viewpoint, wherein the subject is captured from a head-mounted camera.

**Outside-in Static Human Pose Estimation** initially regressed directly to 3D pose from images, without intermediate 2D representation [11–15]; [14, 15] considered the use of volumetric heatmaps to utilize 3D features in images on popular outside-in datasets such as Human3.6M. Soon after, many works applied 2D to 3D lifting models [17–20], taking advantage of accurate 2D pose for 3D tasks. Works showed that joint estimation of 2D and 3D poses is advantageous both in supervised and unsupervised settings [34, 35]. Our work leverages the supervised joint estimation of 2D and 3D poses, building on [34]. We demonstrate Ego-STAN's ability to overcome occlusions and generalize to these scenarios as compared to popular 2D HPE (HRNet) [36] and SOTA ego-HPE [21] methods.

**Outside-in Video 3D Human Pose Estimation** utilizes the temporal information of video to improve 3D HPE [29, 37–42]. More recently, these include the use of deep learning-based sequential models such as Long Short-Term Memory (LSTM) [43] and spatio-temporal relations via temporal Convolutional Neural Networks (CNN) [44]. Enforcing temporal consistency using bone length and direction has been proposed for HPE [45]. More recently, Transformer-based attention mechanisms have gained popularity for factoring in frame significance and receptor-field dependency [26–28, 46]. Others proposed explicit visibility guidance by pseudo-labeling or human labels [29–31].

**Transformers for 3D Pose Estimation** have shown remarkable success in a number of application areas [32], including computer vision via the introduction of Vision Transformers (ViT) [47]. Recent efforts focus on combining CNNs and self-attention mechanisms to reduce parameters and allow for lightweight networks for vision applications [48]. In representing temporal phenomena, Transformers have made their way into many different spatio-temporal tasks [49–52]. For example, in action recognition, [53] fully utilizes Transformers for feature extraction, while in video object segmentation, [54] extracts features with a CNN backbone. For outside-in HPE, PoseFormer [25] utilized a spatio-temporal sequence of 2D keypoints from an off-the-shelf 2D pose estimator to predict the 3D pose. Unlike PoseFormer, the distorted egocentric views preclude us from using such off-the-shelf methods. Ego-STAN addresses this challenge through the supervised 2D heatmap estimation.

**Egocentric Human Pose Estimation.** The $Mo^2Cap^2$ dataset was one of the first large HPE synthetic, single-frame datasets with a cap-mounted fish-eye egocentric camera with static views in the train set and sequences in the test [22]. While Ego-STAN primarily relies on spatio-temporal information in sequential (multi-frame) inputs, it yields competitive results with respect to SOTA on the $Mo^2Cap^2$ single-frame dataset, demonstrating the effectiveness of direct 3D regression over an autoencoder-based model [21]. More recently, the xR-EgoPose dataset, a sequential ego-views dataset, was released, offering a larger (and more realistic) dataset [21]. The work also proposed a single and a dual-branch auto-encoder structure for 3D ego-HPE. Focusing on the fish-eye distortion, [24] use camera parameters in training. Next, to address the depth ambiguity and temporal instability in egocentric HPE, GlobalPose [23] proposed a sequential variational auto-encoder-based model, that uses [21] as a submodule. Since Ego-STAN accomplishes significant improvements over [21] by leveraging spatio-temporal information, it can also be used with GlobalPose [23].

## 3 EGO-STAN

We now develop the proposed **Ego**centric **S**patio-**T**emporal Self-**A**ttention **N**etwork (Ego-STAN) model, shown in Fig 2, which jointly addresses the self-occlusion and the distortion introduced by the ego-centric views. In doing so, we also conduct an in-depth analysis of the relationship between the 2D heatmap and 3D pose estimation. Ego-STAN consists of four modules. Of these, the **feature extraction** and **spatio-temporal Transformer** modules aim to address the self-occlusion problem by regressing information from multiple time steps, while the **heatmap reconstruction** and **3D pose estimator** modules accomplish uncertainty saturation with lighter 2D-to-3D lifting architectures.

### 3.1 FEATURE EXTRACTION

The feature extraction module in Fig. 2 extracts feature maps that identify regions of interest from ego-centric images via multiple non-linear convolutional filters. Building on a ResNet-101 [33]

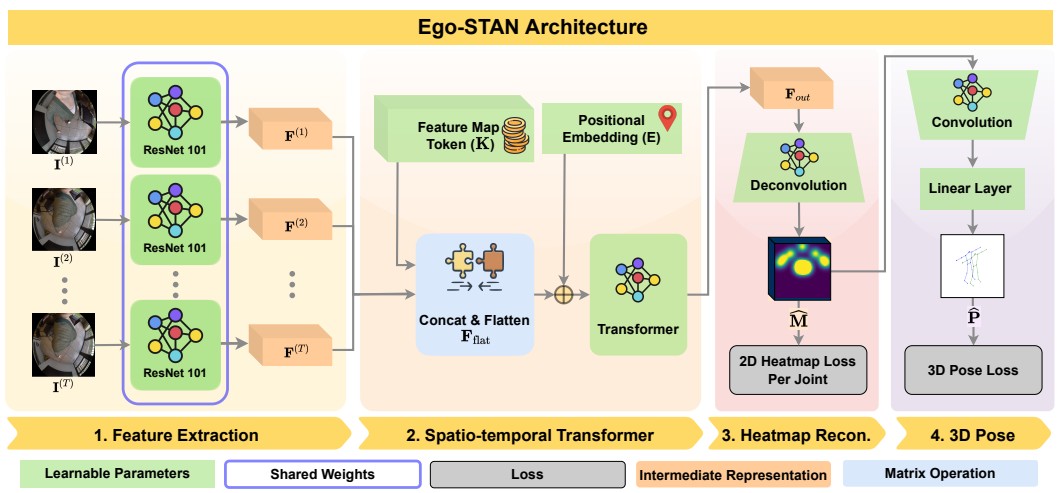

**Figure 2: Ego-STAN Overview**. The proposed Ego-STAN model captures the dynamics of human motion in ego-centric images using Transformer-based spatio-temporal modeling. Ego-STAN uses ResNet-101 as a feature extractor. The proposed Transformer architecture leverages *feature map token* to facilitate spatio-temporal attention to semantically rich feature maps. Our heatmap reconstruction module estimates the 2D heatmap using deconvolutions, which are used by the 3D pose estimator to estimate the 3D joint coordinates.

backbone for extracting image-level features, we introduce a specialized set of learnable parameters – *feature map token* (FMT) – utilized by our Transformer to draw valuable pose information across time-steps. By combining information from different time-steps, Ego-STAN accomplishes 2D heatmap estimation even in challenging cases where views suffer from extreme occlusions, as follows.

**ResNet-101.** Ego-STAN leverages the intermediate ResNet-101 representations to form image-level feature maps. Let $R(\cdot)$ represent ResNet-101's non-linear function that extracts a feature map from a given image $\mathbf{I} \in \mathbb{R}^{H \times W \times C}$ of height $H$, width $W$ and channels $C$. Then, given an image sequence $\mathbb{I}_T = \{\mathbf{I}^{(1)}, \mathbf{I}^{(2)}, ..., \mathbf{I}^{(T)}\}$ of length $T$, where $\mathbf{I}^{(t)}$ is an image at time $t$, we obtain a sequence of feature maps $\mathbb{F}_T = \{\mathbf{F}^{(1)}, \mathbf{F}^{(2)}, ..., \mathbf{F}^{(T)}\}$ by applying $R(\cdot)$ to each image to form $\mathbf{F}^{(t)} \in \mathbb{R}^{\tilde{H} \times \tilde{W} \times \tilde{C}}$ as

$$\mathbf{F}^{(t)} = R(\mathbf{I}^{(t)}). \tag{1}$$

**Feature map token.** To leverage information from past frames to counter occlusions, we require a way to aggregate input feature maps over different times-steps. Specifically, dynamic aggregation to address variable magnitudes of occlusions over frames. To this end, we propose learnable parameters – feature map token (FMT) $\mathbf{K} \in \mathbb{R}^{\tilde{H} \times \tilde{W} \times \tilde{C}}$ – which learns where to pay attention for feature aggregation in conjunction with self-attention [32]. FMT are related to recent works which introduce learnable parameters for classification or *classification tokens*[55] with some key differences which make them a powerful way to aggregate information. As shown in Fig. 3, while a classification token is a single unit token that computes a weighted sum of the feature representations specifically for classification, our proposed feature map token has multiple feature map points, each of which can aggregate from all semantic tokens that are distributed spatially and temporally based on the attention weights, corresponding to a particular location in an image for intermediate 2D heatmap representation. As a result, each unit of the FMT $\mathbf{K}$ learns how to represent accurate semantic features for the heatmap reconstruction module. Furthermore, the cor-

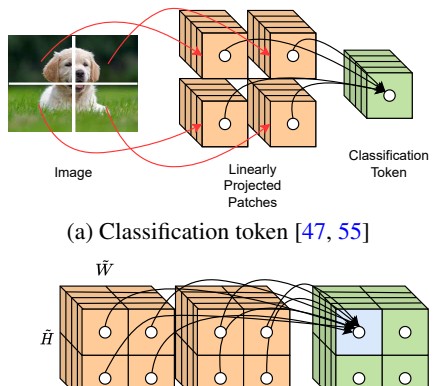

(a) Classification token [47, 55]

(b) Feature map token (this work).

Figure 3: Difference between classification token (top) and feature map token (FMT) (bottom). Each unit of FMT (blue) corresponds to a section of an image for processing, paying attention to input tokens $\mathbf{F}^{(t)}$.

responding attention matrix can be visualized for interpretability as shown in Fig. 1. We randomly initialize the token, $\mathbf{K}$, and concatenate it with the feature map sequence $\{\mathbf{F}^{(t)}\}_{t=1}^{T}$, denoted by

$\texttt{Concatenate}(\cdot)$ along the $\bar{W}$ dimension to obtain $\mathbf{F}_{\text{concat}} \in \mathbb{R}^{\tilde{H} \times \bar{W}(T+1) \times \bar{C}}$ as

$$\mathbf{F}_{\text{concat}} := \texttt{Concatenate}(\mathbf{K}, \mathbb{F}_T) = [\mathbf{K}, \mathbf{F}^{(1)}, \mathbf{F}^{(2)}, ..., \mathbf{F}^{(T)}]. \qquad (2)$$

We flatten the non-channel dimensions with the $\texttt{Flatten}(\cdot)$ operation (mode-3 fibers [56]) in order to serialize the input for the Transformer module to obtain $\boldsymbol{F}_{\text{flat}} \in \mathbb{R}^{\tilde{H}\tilde{W}(T+1) \times \tilde{C}}$ as

$$\boldsymbol{F}_{\text{flat}} := \texttt{Flatten}(\mathbf{F}_{\text{concat}}). \qquad (3)$$

## 3.2 SPATIO-TEMPORAL ATTENTION USING FEATURE MAP TOKEN

Now that we have $\boldsymbol{F}_{\text{flat}}$, that contains both feature maps from multiple time steps and the feature map token, we are ready for spatio-temporal learning. Self-attention learns to map the pairwise relationship between *input tokens* $\boldsymbol{F}_{\text{flat}[r,:]}$ for $r = \{1, \dots, \tilde{H}\tilde{W}(T+1))\}$. This is especially important because it allows the feature map token $\mathbf{K}$ (the first $\tilde{H}\tilde{W}$ rows in $\mathbf{F}_{\text{flat}}$) to look across all of the input tokens in the spatio-temporally distributed sequences to learn where to pay attention.

**Positional Embedding.** Transformer networks need to be provided with additional information about the relative position of input tokens [32]. As our input space often has repetitive background or body positions, it is important to inject positional guidance in order for the network to be able to distinguish identical input tokens. To accomplish this, we add a learnable position embedding $\boldsymbol{E} \in \mathbb{R}^{\tilde{H}\tilde{W}(T+1) \times \tilde{C}}$ element-wise to $\boldsymbol{F}_{\text{flat}}$ to form $\boldsymbol{F}_{\text{pe}} \in \mathbb{R}^{\tilde{H}\tilde{W}(T+1) \times \tilde{C}}$ as

$$\boldsymbol{F}_{\text{pe}} = \boldsymbol{F}_{\text{flat}} + \boldsymbol{E}. \qquad (4)$$

**Self-attention with FMT.** Our Transformer module – $\texttt{Transformer}(\cdot)$ – encodes spatio-temporal information in feature map $\boldsymbol{F}_{\text{pe}}$ with self-attention and returns $\boldsymbol{F}_{\text{tfm}} \in \mathbb{R}^{\tilde{H}\tilde{W}(T+1) \times \tilde{C}}$. Ego-STAN learns FMT weights, $\mathbf{K}$, and the linear projections of the Transformer encoder [32] to understand which tokens are important in the sequence via a hybrid CNN backbones and Transformers motivated from [48, 54]. In the self-attention module, there are three sets of learnable parameters (implemented as a linear layer) that enable this dynamic aggregation via FMT $\boldsymbol{L}_{\text{q}} \in \mathbb{R}^{\tilde{C} \times D}$, $\boldsymbol{L}_{\text{r}} \in \mathbb{R}^{\tilde{C} \times D}$, and $\boldsymbol{L}_{\text{v}} \in \mathbb{R}^{\tilde{C} \times D}$, which are used to form query $\boldsymbol{Q}$, key $\boldsymbol{R}$, and value $\boldsymbol{V}$ for the Transformer module as

$$\boldsymbol{Q} := \boldsymbol{F}_{\text{pe}} \boldsymbol{L}_{\text{q}}, \qquad \boldsymbol{R} := \boldsymbol{F}_{\text{pe}} \boldsymbol{L}_{\text{r}}, \qquad \boldsymbol{V} := \boldsymbol{F}_{\text{pe}} \boldsymbol{L}_{\text{v}}. \qquad (5)$$

Given these matrices, the attention matrix $\boldsymbol{A} \in \mathbb{R}^{\tilde{H}\tilde{W}(T+1) \times \tilde{H}\tilde{W}(T+1)}$ is computed as

$$\boldsymbol{A} := \texttt{Softmax}(\boldsymbol{Q}\boldsymbol{R}^{\top}), \qquad (6)$$

and the subsequent aggregation $\boldsymbol{A}_{\text{v}}$ using the value matrix $\boldsymbol{V}$ as

$$\boldsymbol{A}_{\text{v}} := \boldsymbol{A}\boldsymbol{V}. \qquad (7)$$

Finally, $\boldsymbol{A}_{\text{v}}$ is passed through the feed forward block to form $\boldsymbol{F}_{\text{tfm}}$. These three learnable parameters can therefore dynamically determine the aggregation weights depending on the semantics of the feature maps; can be on independent interest in application that require aggregation of semantics from the feature maps that are distributed spatio-temporally. Note that this aggregation is for a single head in a multi-head attention module. Finally, the action of our Transformer module can be represented as

$$\boldsymbol{F}_{\text{tfm}} := \texttt{Transformer}(\boldsymbol{F}_{\text{pe}}) \text{ or alternatively } \boldsymbol{F}_{\text{tfm}} := \texttt{FeedForward}(\boldsymbol{A}_{\text{v}}). \qquad (8)$$

We only take the first $\tilde{H}\tilde{W}$ tokens corresponding to the feature map token $\mathbf{K}$ from $\boldsymbol{F}_{\text{tfm}}$ and reshape into a $\tilde{H} \times \tilde{W} \times \tilde{C}$ tensor to form the spatio-temporal Transformer output $\mathbf{F}_{\text{out}} \in \mathbb{R}^{\tilde{H} \times \tilde{W} \times \tilde{C}}$ as

$$\mathbf{F}_{\text{out}} := \texttt{Reshape}(\boldsymbol{F}_{\text{tfm}:\tilde{H}\tilde{W},:}). \qquad (9)$$

As a result, these modules, and specifically the feature map token, create an accurate semantic map for heatmap reconstruction (further discussed in Sec. 3.3).

**Slice and average variant.** To explore the impact of FMT, we compare with two spatio-temporal model variants without FMT. The first variant is called *slice*. Since we are interested in estimating the 3D pose of the current frame from given a past frame sequence, we take the indices of the tokens that are respective to the current frame in the token sequence. Given a sequence of tokens (without FMT), we take the last $\tilde{H}\tilde{W}$ indices from $\boldsymbol{F}_{\text{tfm}} \in \mathbb{R}^{\tilde{H}\tilde{W}T \times \tilde{C}}$ to be deconvolved. Formally we have:

$$\mathbf{F}_{\text{out-slice}} := \texttt{Reshape}(\boldsymbol{F}_{\text{tfm} -\tilde{H}\tilde{W}:,:}). \qquad (10)$$

The *avg* variant reduces the spatial dimension by averaging over spatially same but temporally different tokens. Specifically, we take $\boldsymbol{F}_{\text{tfm}} \in \mathbb{R}^{\tilde{H}\tilde{W}T \times \tilde{C}}$ from (8) and average over the $T$ dimension,

$$\mathbf{F}_{\text{out-avg}} := \texttt{Average}(\boldsymbol{F}_{\text{tfm}}, \dim = T). \qquad (11)$$

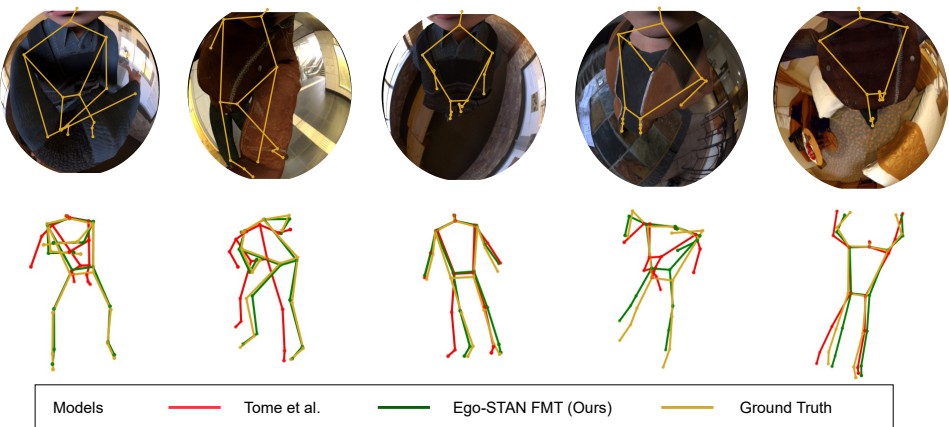

Figure 4: **Qualitative evaluation on highly occluded frames**. We demonstrate the qualitative performance of Ego-STAN with feature map token (FMT), compared with the SOTA dual-branch model [21] on self-occluded frames. The top row shows the frames superimposed with the ground truth 2D joint location skeleton (in gray). We observe that Ego-STAN is significantly more robust to occlusions relative to the dual-branch model [21].

### 3.3 HEATMAP RECONSTRUCTION

**Feature map to heatmap.** Our goal is to leverage deconvolution layers to reconstruct ground truth 2D heatmaps of time $T$, $\mathbf{M} \in \mathbb{R}^{h \times w \times J}$, of height and width, $h \times w$, for each major joint in the human body ($J$). To this end, $\mathbf{F}_{\text{out}}$ is passed through two deconvolution layers to estimate $\widehat{\mathbf{M}} = \in \mathbb{R}^{h \times w \times J}$ as

$$\widehat{\mathbf{M}} := \texttt{Deconv}(\mathbf{F}_{\text{out}}), \tag{12}$$

trained via a mean square error, $\texttt{MSE}(\cdot)$-based loss $\mathcal{L}_{2D}$:

$$\mathcal{L}_{2D}(\mathbf{M}, \widehat{\mathbf{M}}) = \texttt{MSE}(\mathbf{M}, \widehat{\mathbf{M}}). \tag{13}$$

### 3.4 3D POSE ESTIMATION

**Heatmap to pose.** We leverage a simple convolution block followed by linear layers to lift the 2D heatmaps to 3D poses. As opposed to the SOTA egocentric pose estimator [21], which uses a dual branched auto-encoder structure aimed at preserving the uncertainty information from 2D heatmaps, we (somewhat surprisingly) find that such a complex auto-encoder design is in fact not required, and our simple architecture accomplishes this task more accurately (see section 4.1). Therefore, given the predicted heatmap $\widehat{\mathbf{M}}$, we predict the 3D coordinates of the joints $\widehat{\boldsymbol{P}} \in \mathbb{R}^{J \times 3}$ at time $T$ as

$$\widehat{\boldsymbol{P}} := \texttt{Linear}\big(\texttt{Convolution}(\widehat{\mathbf{M}})\big). \tag{14}$$

To estimate the 3D pose using the reconstructed 2D heatmaps (13), we use three different types of loss functions – i) squared $\ell_2$-error $\mathcal{L}_{\ell_2}(\cdot)$, ii) cosine similarity $\mathcal{L}_\theta(\cdot)$, and iii) $\ell_1$-error $\mathcal{L}_{\ell_1}(\cdot)$ between $\widehat{\boldsymbol{P}}$ and $\boldsymbol{P}$. These loss functions impose the closeness between $\boldsymbol{P}$ and $\widehat{\boldsymbol{P}}$ in multiple ways. As compared to [21], our $\ell_1$-norm promotes the solutions to be robust to outliers [57], as corroborated by our ablations in section 4.2. As a result, our 3D loss for regularization parameters $\lambda_\theta$ and $\lambda_{\ell_1}$ is

$$\mathcal{L}_{3D}(\boldsymbol{P}, \widehat{\boldsymbol{P}}) = \mathcal{L}_{\ell_2}(\boldsymbol{P}, \widehat{\boldsymbol{P}}) + \lambda_\theta \mathcal{L}_\theta(\boldsymbol{P}, \widehat{\boldsymbol{P}}) + \lambda_{\ell_1} \mathcal{L}_{\ell_1}(\boldsymbol{P}, \widehat{\boldsymbol{P}}) \text{ where,} \tag{15}$$

$$\mathcal{L}_{\ell_2}(\boldsymbol{P}, \widehat{\boldsymbol{P}}) := \|\widehat{\boldsymbol{P}} - \boldsymbol{P}\|_2^2, \ \mathcal{L}_\theta(\boldsymbol{P}, \widehat{\boldsymbol{P}}) := \sum_{i=1}^J \frac{\langle \boldsymbol{P}_i, \widehat{\boldsymbol{P}}_i \rangle}{\|\boldsymbol{P}_i\|_2 \|\widehat{\boldsymbol{P}}_i\|_2}, \ \text{and} \ \mathcal{L}_{\ell_1}(\boldsymbol{P}, \widehat{\boldsymbol{P}}) := \sum_{i=1}^J \|\widehat{\boldsymbol{P}}_i - \boldsymbol{P}_i\|_1.$$

Thus, the overall loss function to train Ego-STAN comprises of the 2D heatmap reconstruction loss and the 3D loss, as shown in (13) and (15), respectively.

## 4 EXPERIMENTS

We now analyze the performance of Ego-STAN as compared to the SOTA ego-HPE methods. Additionally, we carry-out a systematic analysis of the incremental contributions by each component of Ego-STAN via extensive ablations. We analyze the performance on the xR-EgoPose dataset [21], the only dataset with a sequential ego-view training set for detailed ablations and analysis via the

Table 1: Comparative quantitative evaluation of Ego-STAN against the SOTA Ego-HPE methods. Proposed Ego-STAN variants have the highest accuracies across the nine actions with the feature map token (FMT) variant having the lowest overall MPJPE (lower is better); our results are averaged over three random seeds.

| Approach | Evaluation error (mm) | Game | Gest. | Greet | Lower Stretch | Pat | React | Talk | Upper Stretch | Walk | All |
|---|---|---|---|---|---|---|---|---|---|---|---|
| Martinez et. al. [19] | Upper body | 58.5 | 66.7 | 54.8 | 70.0 | 59.3 | 77.8 | 54.1 | 89.7 | 74.1 | 79.4 |
| | Lower body | 160.7 | 144.1 | 183.7 | 181.7 | 126.7 | 161.2 | 168.1 | 159.4 | 186.9 | 164.8 |
| | Average | 109.6 | 105.4 | 119.3 | 125.8 | 93.0 | 119.7 | 111.1 | 124.5 | 130.5 | 122.1 |
| Tome et. al. [21] single-branch | Upper body | 114.4 | 106.7 | 99.3 | 90.0 | 99.1 | 147.5 | 95.1 | 119.0 | 104.3 | 112.5 |
| | Lower body | 162.2 | 110.2 | 101.2 | 175.6 | 136.6 | 203.6 | 91.9 | 139.9 | 159.0 | 148.3 |
| | Average | 138.3 | 108.5 | 100.3 | 133.3 | 117.8 | 175.6 | 93.5 | 129.0 | 131.9 | 130.4 |
| Tome et. al. [21] dual-branch | Upper body | 48.8 | 50.0 | 43.0 | 36.8 | 48.6 | 56.4 | 42.8 | 49.3 | 43.2 | 50.5 |
| | Lower body | 65.1 | 50.4 | 46.1 | 65.2 | 70.2 | 65.2 | 45.0 | 58.8 | 72.2 | 65.9 |
| | Average | 56.0 | 50.2 | 44.6 | 51.5 | 59.4 | 60.8 | 43.9 | 53.9 | 57.7 | 58.2 |
| Zhang et. al. [24] | Upper body | - | - | - | - | - | - | - | - | - | - |
| | Lower body | - | - | - | - | - | - | - | - | - | - |
| | Average | 36.8 | 34.1 | 36.7 | 50.1 | 57.2 | 34.4 | 32.8 | 54.3 | 52.6 | 50.0 |
| **Ego-STAN Slice (Ours)** | Upper body | 27.2 | 30.0 | 36.3 | 24.0 | 21.3 | 25.4 | 25.3 | 34.2 | 25.5 | 30.2 |
| | Lower body | 38.5 | **30.9** | **33.2** | 54.5 | **32.1** | 35.6 | **29.5** | 64.0 | **55.9** | 55.5 |
| | Average | 32.9 | 30.4 | 34.8 | 39.2 | **26.7** | 30.5 | **27.4** | 49.1 | **40.7** | 42.8 |
| **Ego-STAN Avg. (Ours)** | Upper body | **25.4** | **26.7** | **31.2** | 25.9 | **20.7** | **23.3** | 23.9 | 33.7 | 26.7 | 29.9 |
| | Lower body | **38.1** | 32.7 | 35.0 | 54.7 | 34.6 | **34.3** | 31.2 | 61.2 | 57.2 | 54.3 |
| | Average | **31.7** | **29.7** | **33.1** | 40.3 | 27.7 | **28.8** | 27.5 | 47.4 | 42.0 | 42.1 |
| **Ego-STAN FMT (Ours)** | Upper body | 25.8 | 28.7 | 35.4 | **23.4** | 22.6 | 24.1 | 25.9 | **30.9** | 25.2 | **28.2** |
| | Lower body | 40.3 | 34.5 | 38.3 | **54.4** | 35.9 | 35.0 | 33.4 | **57.6** | 56.5 | **52.6** |
| | Average | 33.1 | 31.6 | 36.9 | **38.9** | 29.2 | 29.6 | 29.7 | **44.3** | 40.9 | **40.4** |

Mean Per-Joint Position Error (MPJPE) metric, to the best of our knowledge. In addition, we also evaluate on Human3.6M [16], an outside-in sequential real-world 3D HPE dataset, and on Mo$^2$Cap$^2$ [22], an ego-HPE dataset with static synthetic train set and real sequential test using MPJPE, to analyze generalization, and adaptability with other pose estimation backbones. On Human 3.6M, we compare the results with and without Ego-STAN on a popular outside-in HPE method [36] and also against the SOTA ego-HPE model [21]. Here, in addition to MPJPE, we also report the Percentage of Correct Keypoint (PCK), a popular metric for Human3.6M, to gauge 2D joint estimation accuracy. Since 3D HPE crucially depends on accurate heatmap (2D) estimation, PCK reveals the capabilities of learned representations. Our results report the average performance across three random seeds; details to allow *reproducibility and the code* are listed in A.2.4 and in the supplementary materials.

## 4.1 RESULTS

**Results (xR-EgoPose).** Tab. 1 shows the MPJPE achieved by Ego-STAN and its variants on the xR-EgoPose test set, as compared to SOTA ego-HPE models [21] (a dual-branch autoencoder model, and its single branch variant), a popular outside-in baseline [19], and [24]. For fair comparison, since [24] requires camera parameters for training, we compare against the dual-branch model of [21]. Ego-STAN variants perform the best across different actions and individual joints, as shown in Tab. 1 and Fig. 7 in A.1, respectively, with Ego-STAN FMT achieving the best average performance. Ego-STAN FMT outperforms the dual-branch model proposed in [21] by a substantial **17.8 mm (30.6%)**, averaged over all actions and joints (Tab. 1). Remarkably, across joints in Fig. 7, **Ego-STAN FMT shows an improvement of 40.9 mm (39.4%) on joints with the highest error in the SOTA [21], with an average improvement of 35.6 mm (38.2%)** over these (left hand, right hand, left foot, right foot, left toe base, and right toe base) joints. Ego-STAN FMT is also most robust to occlusions evident from the lower and upper stretching actions, which suffer from heaviest occlusions (Fig. 7(b)). This robustness is also exhibited by Ego-STAN variants in the violin plots shown in Tab. 3. For a qualitative comparison, we show the estimation results on a few highly self-occluded frames in Fig. 4, further demonstrating the superior properties of Ego-STAN FMT over the SOTA ego-HPE methods.

**Results (Mo$^2$Cap$^2$).** Since Mo$^2$Cap$^2$ contains a static train set, this allows us to analyze the impact of direct 3D regression. Here, Ego-STAN improves the MPJPE by 10% on the Mo$^2$Cap$^2$ test set over the SOTA [21]. Results and additional details are shown in Sec. A.2.3 and Tab. 7.

Table 2: Quantitative evaluation on Human3.6M for HPE. Accuracy of both 2D HPE and 3D HPE are improved with Ego-STAN even under high occlusions; here Sld: Shoulder, Elb: Elbow.

| Approach (2D, PCK@0.05, ↑) | Sld | Elb | Wrist | Hip | Knee | Ankle | Spine | All |
|---|---|---|---|---|---|---|---|---|
| Sun [36] | 0.763 | 0.761 | 0.713 | 0.807 | 0.916 | 0.921 | 0.900 | 0.847 |
| Sun [36] + **Ego-STAN** | **0.941** | **0.851** | **0.781** | **0.918** | **0.923** | **0.933** | **0.950** | **0.912** |
| **Approach (3D, MPJPE(mm), ↓)** | **Sld** | **Elb** | **Wrist** | **Hip** | **Knee** | **Ankle** | **Spine** | **All** |
| Tome [21] Protocol 1 | 131.7 | 172.9 | 209.1 | 42.0 | 125.9 | 178.8 | 74.2 | 119.4 |
| **Ego-STAN (ours)** Protocol 1 | **122.5** | **163.5** | **198.4** | **30.4** | **95.8** | **125.6** | **63.5** | **109.3** |
| Tome [21] Protocol 2 | 51.0 | 113.5 | 134.5 | 67.8 | 84.3 | 108.7 | **43.4** | 73.8 |
| **Ego-STAN (ours)** Protocol 2 | **40.6** | **94.0** | **128.2** | **76.0** | **70.0** | **89.4** | 44.8 | **68.9** |

| Models | —— HRNet | —— Ego-STAN FMT (Ours) | —— Ground Truth |

(a) Directions         (b) Taking Photograph         (c) Sitting

Figure 5: **Qualitative evaluation on Human3.6M dataset.** We demonstrate the qualitative performance of Ego-STAN on occluded frames of Human3.6M. Compared to a popular static 2D outside-in HPE method [36], Ego-STAN better estimates occluded joints (highlighted with light blue box). Video attached to sup. material.

**Results (Human3.6M).** Outside-in views do not suffer from the same level of self-occlusions and distortions. As a result, our results highlight Ego-STAN's ability to leverage spatio-temporal information via FMT, of independent interest for HPE in-general. As demonstrated in Tab. 2, wrapping Ego-STAN on a 2D HPE backbone improves the PCK by 8%, underscoring its adaptability and its ability to generalize to real-world data. Moreover, the improvements of 9% on Protocol 1 and 7% on Protocol 2 against the SOTA egocentric HPE [21] strengthens the point that Ego-STAN can be used for real-world data. Additional details are presented in Sec. A.2.2.

## 4.2 ABLATION STUDIES

We perform a series of ablation studies on xR-EgoPose to analyze the incremental effect of each element of Ego-STAN. We begin by presenting short descriptions of these elements. Here, $+$ represents the addition of certain element and $\Delta$ indicates replacing an element with another.

- **Baseline.** Reproduced model [21] trained by $\mathcal{L}_{2D}(\mathbf{M}, \widehat{\mathbf{M}})$ (13), $\mathcal{L}_{\ell_2}(\boldsymbol{P}, \widehat{\boldsymbol{P}})$, & $\mathcal{L}_{\theta}(\boldsymbol{P}, \widehat{\boldsymbol{P}})$ (14).

- $+\ \ell_1-$**norm.** Above **Baseline** with the addition of $\mathcal{L}_{\ell_1}(\boldsymbol{P}, \widehat{\boldsymbol{P}})$ in the cost function in Sec. 3.3.

- $+$ **Temporal TFM.** Temporal Transformer (TFM) which attends to the sequence of latent vectors produced by the autoencoder structure in the **Baseline** $+\ \ell_1-$**norm**.

- $\Delta$ **Direct 3D Regression.** Replaces the dual branch autoencoder and the Temporal TFM with a simple neural network to directly regress to 3D pose from heatmaps; see Sec. 3.4,

- $+$ **Spatial-only TFM.** Addition of self-attention on the feature map generated by a single frame.

- $+$ **Ego-STAN w/ Slice.** Addition of temporal attention leads to Ego-STAN. This variant of Ego-STAN uses sliced tokens of the current frame (10).

- $\Delta$ **Ego-STAN w/ avg.** Replaces slicing with token averaging across the $T$ dimension (11).

- $\Delta$ **Ego-STAN w/ FMT.** Our main proposed method, which replaces averaging with FMT (2).

Fig. 7, Tab. 3, and Fig. 6, show the performance of each incremental model, illustrating each effect on the overall performance of Ego-STAN averaged across three random seeds. We observe the following.

*Where we employ temporal attention matters.* Temporal attention on the feature map sequence yields better performance than on the latent vector sequence arising from autoencoder structure (Temporal TFM vs. Ego-STAN variants). This demonstrates that 2D heatmap-based representations are adequate for HPE, and autoencoders may create unnecessary information bottlenecks.

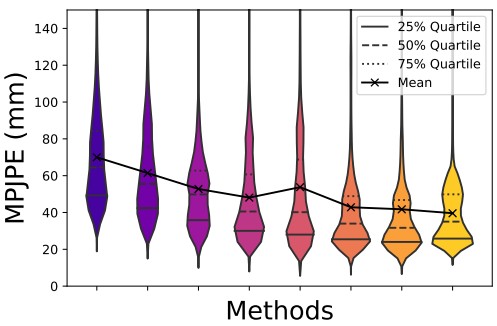

Figure 6: **Overall MPJPE analysis across different methods.** The violin plots demonstrate the contribution of each component of Ego-STAN for seed 42. Ego-STAN and its variants exhibit superior mean and variance properties (colors correspond to legend shown in Tab. 3).

Table 3: **Overview of ablations.** From top to bottom, $+$ and $\Delta$ denote cumulative and change via replacement, respectively. MPJPE for each model is reported with sample standard deviation from 3 different seeds.

| Legend | Method | Parameters (Millions) | MPJPE (mm) |
|---|---|---|---|
| | **Baseline** (Tome et. al. [21]) | 141 | $65.7 \pm 4.0$ |
| | $+ \ell_1$-norm | 141 | $60.1 \pm 3.1$ |
| | $+$ Temporal TFM | 141 | $55.7 \pm 2.7$ |
| | $\Delta$ Direct 3D Regression | 101 | $50.8 \pm 1.7$ |
| | $+$ Spatial-only TFM | 109 | $52.5 \pm 3.0$ |
| | $+$ Ego-STAN w/ Slice | 110 | $42.8 \pm 0.0$ |
| | $\Delta$ Ego-STAN w/ Avg. | 110 | $42.1 \pm 2.1$ |
| | $\Delta$ Ego-STAN w/ FMT | 110 | $40.4 \pm 0.1$ |

*Direct 3D regression works better than auto-encoder structure(s)* indicating that as opposed to the conjecture in SOTA [21], the uncertainty information is effectively captured by the 2D heatmaps obviating the need for an autoencoder structure. Direct 3D regression can also be viewed as a variant of [24] without extra information about the camera parameters. We hypothesize that replacing the autoencoder structure may also be the primary source of improvements reported in [24] for the static case. This is encouraging since camera information may be impractical to obtain in the real world.

*Spatio-temporal information is essential.* From Tab. 3 we note a slight performance dip (increased MPJPE) when only spatial attention is used. This indicates that for a static setting, using raw feature maps is better than using spatial attention. Moreover, incorporating a temporal aspect significantly improves the performance, underscoring its role in Ego-STAN variants. Further improvement due to FMT demonstrates that how we choose to aggregate information from feature maps matters.

*Reducing trainable parameters.* Ego-STAN variants lead to a **reduction of 31M (22%) trainable parameters as compared to the SOTA [21]**. This is attributed to our hybrid architecture which a) replaces the auto-encoder with direct 3D regression module (-28%), and b) leverages a FMT-based Transformer encoder-only module (+6%) obviating the need for a decoder [54]. These findings are in line with recent works which show improvements with CNN-Transformer hybrids [48].

*Consistent and accurate HPE.* Finally, in Fig. 6 we observe that as we progress to the right, in addition to the reduction in the overall MPJPE, the error distribution becomes lower and more consistent, indicating better variance properties (shorter vertically and wider at the bottom). This robustness can also be attributed to our $\ell_1$-based 3D-loss (15).

Overall, our results demonstrate that Ego-STAN effectively handles distortions and self-occlusions.

## 5 DISCUSSION

**Summary.** Ego-HPE is challenging due to self-occlusions and distorted views. To address these challenges, we design a spatio-temporal hybrid architecture which leverages CNNs and Transformers using learnable parameters (FMT) that accomplish spatio-temporal attention, significantly reducing the errors caused by self-occlusion, especially in joints which suffer from high error in SOTA works. Our proposed model(s) – Ego-STAN – accomplishes consistent and accurate ego-HPE and HPE in general, while notably reducing the number of trainable parameters, making it suitable for cutting-edge full body motion tracking applications such as activity recognition, surgical training and immersive $x$R applications. This resulting transformer makes foundational contributions to spatio-temporal data analysis, impacting advances in ego-pose estimation and beyond.

**Limitations, and future work.** Although Ego-STAN demonstrates generalization capabilities on outside-in HPE datasets, there are no real-world ego-HPE sequential datasets. And while such datasets are developed, our future efforts will focus on developing transfer learning-based models which can work under domain shifts and variations in camera positions. This will lead to robust HPE models which can adapt to a variety of environments for real world critical applications.

ETHICS STATEMENT

Human pose estimation applications include surveillance by public or private entities, which raises privacy invasion and human rights concerns. There is a need to educate practitioners and the users of applications relying on such technologies about such potential risks. Research on privacy preserving machine learning offers a way to mitigate these risks. Simultaneously, there is also a need to provide more legal protections for users and their data, and regulations for entities utilizing this data.

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
