# OpenReview forum: "Spatio-temporal Self-Attention for Egocentric 3D Pose Estimation"
_ICLR.cc/2023/Conference — Submitted to ICLR 2023_

### Official Review · Reviewer_cA4Q · 2022-10-24

**Confidence:** 4
**Correctness:** 3
**Technical Novelty And Significance:** 3
**Empirical Novelty And Significance:** Not applicable
**Recommendation:** 6

**Clarity, Quality, Novelty And Reproducibility:**

The paper is almost well written and easy to follow, with minor details missing as described above.
Implementation details and code are provided in the supplementary material, which eables reproducibility of the paper.

**Strength And Weaknesses:**

Strength:
1) A feature map token (FMT) with learnable parameters is designed to aggregate information from multiple image frames to augment the contextual feature embedding of a spatial-temporal Transformer.
2) A hybrid spatial-temporal Transformer is proposed to process convolutional feature maps of image sequence with spatial-temporal self-attention.
3) Extensive experiments are conducted to verify the proposed method. Superior performance is achieved with the proposed Ego-STAN compared with related SOTA methods in egocentric HPE. Detailed ablation studies are also conducted to analyze the impact of each component of Ego-STAN.

Weakness:
1) Most of the performance improvement seems to come from the spatial-temporal Transformer architecture (+10%), while the improvement by the core component of FMT is limited (+2%). The spatial-temporal Transformer can hardly be seen as the technical contribution of this work.
2) Some technical details are missing or not clear. The network (probably a fully connected layer) for estimating the feature map token is not given. Is each unit of FMT accompanied with a unique network or a shared network? The details for the learnable position embedding are not provide either. In Figure 2, there should be a connection from CNN feature maps (F) to feature map token (K).

**Summary Of The Paper:**

This paper proposes a spatial-temporal self-attention model (Ego-STAN) for egocentric-view human pose estimation (HPE) task. The main contribution is a learnable feature map token which aggregates features from past image frames which are then used as input to a spatial-temporal Transformer. Experiments on three public datasets (two egocentric-view and one outside-in-view) are conducted to verify the effectiveness of the proposed method.

**Summary Of The Review:**

My overall rating is positive with several concerns. Please see above comments.

---

> ### Author Response · Authors · 2022-11-19
> **Response to Reviewer cA4Q**
>
> Thank you for recognizing the novelty of our work and for your detailed summary and comments. Below we address the two comments. We hope these responses lead to a further positive evaluation of our work, and we will be happy to hear from you if you have any follow-up questions. Thank you for your time and feedback.
>
> > Q1. Most of the performance improvement seems to come from the spatial-temporal Transformer architecture (+10%), while the improvement by the core component of FMT is limited (+2%). The spatial-temporal Transformer can hardly be seen as the technical contribution of this work.
>
> **Response:**
>
> **Leveraging Spatio-temporal Information in EPE:**
> The primary premise of the work is to utilize the spatio-temporal information in ego-views to counter self-occlusions and fish-eye distortions. These unique challenges mean that outside-in pose estimation models are not suitable for the case of EPE (see response to kGU6 Q5.3 for an additional discussion about why outside-in spatio-temporal approaches are not suitable for EPE). The spatio-temporal Transformer here is only a means of feature extraction which is scalable in the future to any other spatio-temporal model. Therefore, the key fundamental contribution is to leverage spatio-temporal information to counter severe self-occlusions and distortions.
>
> Moreover, designing an appropriate way to use spatio-temporal attention is extremely non-trivial for 3D EPE since this involves careful conditional optimization (3D predictions relying on predicted 2D heatmaps). Next, aggregating information across frames is also challenging, here we develop slice and averaging then finally design FMT that learns how/where to pay attention. These performance gains should be evaluated on their potential to impact EPE where works are increasingly relying on SOTA dual-branch model, which we show is complex and ineffective.
>
> Furthermore, our work is also the first to model spatio-temporal dynamics to counter self-occlusions in EPE. This contribution is extremely significant since none of the previous works in EPE address self-occlusions [1,3,4] or leverage spatio-temporal modeling.
>
> > Q2. Some technical details are missing or not clear. The network (probably a fully connected layer) for estimating the feature map token is not given. Is each unit of FMT accompanied with a unique network or a shared network? The details for the learnable position embedding are not provide either. In Figure 2, there should be a connection from CNN feature maps (F) to feature map token (K).
>
> **Response:**
>
> **Clarifications on FMT.** The FMT rely on a shared CNN network. We have updated the discussion to make this clear. We have also added an intuitive explanation of FMT in Appendix A.3.
>
> - **Clarifying FMT & Figure 2.** Figure 2 correctly represents the diagram of our work. The principle of FMT cannot be understood as stand-alone, it must consider the whole spatio-transformer block (sections 3.1 and 3.2). Intuitively, the weights of FMT are updated so that it understands where to pay attention, given a sequence of feature maps from the CNN backbone. In other words, FMT learns how to position its direction of the token vectors so that given a set of feature maps of certain visibility (occlusion), the linear projections $Q$ and $K$ can determine the weight of the attention matrix for aggregation on the past or the current frame.
>
> - **Missing details of positional embeddings.** Thank you for the suggestion. Learnable positional embeddings are well thought-of and explained in original works [13, 14]. We now point interested readers to these works in Appendix A.5.

---

### Official Review · Reviewer_kGU6 · 2022-10-26

**Confidence:** 2
**Correctness:** 3
**Technical Novelty And Significance:** 2
**Empirical Novelty And Significance:** 2
**Recommendation:** 3

**Clarity, Quality, Novelty And Reproducibility:**

Clarity: Some details of the proposed method and experimental setting are not clear or missing.

Quality: The technical novelty of the proposed method is somewhat limited. The experimental evaluation should be enhanced significantly.

Novelty: The problem is not new, the novelty of the proposed method is somewhat limited.

Reproducibility: Authors provided the code in the supplementary material.


**Strength And Weaknesses:**

Strength:

1. The proposed method achieves superior performance on the xR-EgoPose dataset.

2. Comprehensive ablation study is conducted.


Weakness:

1. The technical novelty of the proposed method is somewhat incremental. The proposed feature map token is just a simple way to perform global attention from the extracted features of CNN backbone. Some of the tricks used for performance improvement like L1-based 3D loss are incremental.

2. Although there is a graphical illustration of the FMT, it is still not clear about the specific operation. Concrete equations can be provided to make it clear about the actual operations.

3. There are some details about the overall framework are missing. For example, how many (image) frames are used to reconstruct a 3D pose? Are the previous t frames used to reconstruct the (t+1) 3D pose? These settings should be made clear. Also, in the experiments, are these settings keep the same for all the competing methods?

4. Direct regression from heatmap (or 2D pose) to 3D pose with a simple neural-network based design is not a significant contribution. This is common in the literature of general 3d human pose estimation.

5.  The experimental evaluation is weak. 1) Only one ego-HPE method (i.e., reference [21]) is used for comparison on the Ego-dataset. Reference [21] was published in 2019. It may not represent the state-of-the-art. 2) 3d human pose estimation from image/video is a well-established field with many recent works using vision-transformer for achieving impressive results. It is important to also include the recent state-of-the-art (image-based and video-based) outside-in 3d pose estimation methods for a comprehensive evaluation on the datasets used in the experiments. There are many works also focus on occlusion-robust pose estimation. Otherwise, the result comparison is not convincing, and it is difficult to justify why a specific ego-focused HPE method is needed.

6. How the proposed method can address the occlusion issue? Some results and visualization should be provided.

7. The computational complexity aspect (e.g., FLOPs) of the method should be discussed and compared with sota methods.


**Summary Of The Paper:**

This paper presents spatio-temporal transformer model for ego-HPE. A spatial concept called feature map tokens is introduced to attend to all other spatial units in the spatio-temporal feature maps. The proposed method achieved superior performance on the xR-EgoPose dataset.

**Summary Of The Review:**

Overall, the technical novelty of the proposed method is incremental and the experimental evaluation cannot fully justify the effectiveness of the proposed method without a comprehensive comparison to the sota 3d human pose estimation approaches.

---

> ### Author Response · Authors · 2022-11-19
> **Response to Reviewer kGU6 (Part 3 of 3)**
>
>
> > Q4. Direct regression from heatmap (or 2D pose) to 3D pose with a simple neural-network based design is not a significant contribution. This is common in the literature of general 3d human pose estimation.
>
> **Response:** In addition to the impact of direct regression in Q1, note that given the same CNN backbone, Martinez et al. [12] uses direct regression on 2D keypoints and achieves MPJPE of 122.1mm.  Our work demonstrates that directly regressing from 2D heatmaps with a few CNN layers can achieve MPJPE of 50.8mm instead of the dual-branch structure (MPJPE 58.2mm) proposed in [1].
>
> >Q5. The experimental evaluation is weak. 1) Only one ego-HPE method (i.e., reference [21]) is used for comparison on the Ego-dataset. Reference [21] was published in 2019. It may not represent the state-of-the-art. 2) 3d human pose estimation from image/video is a well-established field with many recent works using vision-transformer for achieving impressive results. It is important to also include the recent state-of-the-art (image-based and video-based) outside-in 3d pose estimation methods for a comprehensive evaluation on the datasets used in the experiments. 3) There are many works also focus on occlusion-robust pose estimation. Otherwise, the result comparison is not convincing, and it is difficult to justify why a specific ego-focused HPE method is needed.
>
>
> **Response Q5.1:** We would like to gently correct the reviewer's understanding. We compare Ego-STAN's performance with **four resnet-based baselines**  in Table 1 which are standard in the literature ([12], single-branch [1], dual-branch [1], and [4]). To the best of our knowledge, [1] indeed is the state-of-the-art in this case, along with [4] which relies on camera intrinsics labels. Ego-STAN outperforms all baselines by a significant margin, and is also the most consistent (Fig. 6).
>
> **Response Q5.2:**  The recent works that utilize sequential transformers [5-8] for outside-in view (Human3.6m dataset) focus on estimating 3D pose from precomputed sequential 2D keypoints by an off-the-shelf model. While these methods aim to predict 3D pose from a sequence of precomputed 2D keypoints, our work aims to predict accurate 3D pose from accurate 2D heatmap priors. More importantly, sequential keypoints-based methods were able to utilize the off-the-shelf models since do not suffer from distortions or nearly as much occlusions, whereas this cannot be taken for granted for EPE since going from image to 2D is in itself a challenging problem. Therefore, outside-in methods [5-8] are not relevant in EPE.
>
> **Response Q5.3:**
>
> The differences between outside-in occlusions and those in ego-views are discussed in the first part of response in Q1. Recent works on outside-in occlusion-robust 3D pose estimation [9-11] have not been compared because a) these are customized for outside-in views, and/or b) require additional information about the joint visibility not available in EPE. For example, [9] requires optical-flow which does not apply for EPE because of the constantly moving background caused by ego-views. Additionally, both [9] and [10] require the labeling of joint visibility with the “Cylinder man” proposed from [9], but this is not applicable for EPE due to the camera distortion and dynamic view. Furthermore, [11] has access to labeled joint visibility while EPE does not.
>
>
> >Q6. How the proposed method can address the occlusion issue? Some results and visualization should be provided.
>
> **Response:** Qualitative visualizations of highly occluded images are provided in Fig. 4 and 5. In addition, videos in supplementary materials cleary demonstrate Ego-STAN's superior performance in achieving low jitter as compared to SOTA. Additionally, Table 4 & 5 show the most improvements on highest error and most occluded joints in SOTA [1].Ego-STAN's temporal feature maps infer occluded joint estimations by spatio-temporal aggregation across frames to produce accurate 3D HPE.
>
>
>
> >Q7. The computational complexity aspect (e.g., FLOPs) of the method should be discussed and compared with sota methods.
>
> **Response:** The number of parameters in each model of our ablation studies is displayed in Table 3. Per reviewer’s request, computational complexity in terms of MACs is included in Appendix A.4. Ego-STAN linearly scales with the number of frames $T$, which is reasonable since Ego-STAN is a spatio-temporal model. Moreover, Ego-STAN also achieves a 22% reduction in parameters as compared to SOTA [1] since there is no autoencoder structure.

---

> ### Author Response · Authors · 2022-11-19
> **Response to Reviewer kGU6 (Part 2 of 3)**
>
>  >Q1. The technical novelty of the proposed method is somewhat incremental. The proposed feature map token is just a simple way to perform global attention from the extracted features of CNN backbone. Some of the tricks used for performance improvement like L1-based 3D loss are incremental.
>
> **Response:**
>
> **Novelty of the Spatio-temporal modeling:** The key contribution of this work is to leverage the spatio-temporal information to address the extreme self-occlusions and fish-eye distortions that plague EPE. As opposed to EPE, the outside-in undistorted view allows far less joint occlusions, making it easier for outside-in approaches to estimate occluded joint locations. Recent spatio-temporal outside-in works focus on regressing the 3D pose from a sequence of 2D keypoints (list of Euclidean joint locations) [5-8], with works also relying on additional joint visibility labels to address occlusions [9-11]. In contrast, the self-occlusions and distortions in our EPE make it extremely challenging to estimate 2D joint locations for 3D HPE, that to without access to visibility labels.
>
> Our work fundamentally addresses these unique challenges of EPE by estimating accurate 2D joint locations via heatmaps (which can model uncertainty in joint locations). Furthermore, Ego-STAN dynamically aggregating the temporal feature maps with FMT to counter the heavily self-occluded frames, accomplishing unprecedented performance gains. Additionally, nearly all other ego-pose methods are purely spatial-based [1,3,4], and therefore our spatio-temporal modeling is novel and a fundamental contribution to EPE.
>
>
> Further novel contributions of the paper include the following:
>
> - **Surprising result on Direct Regression:** Our contribution in this context is to re-introduce direct regression for pose estimation overlooked by recent works on EPE [1,2,9]. Specifically, we demonstrate via extensive ablations [Section 4.2, Table 3] that the dual-branch architecture proposed by SOTA [1] is complex and in fact hurts the pose estimation performance (a first in the literature to the best of our knowledge). Since recent methods [2,9] rely on SOTA’s dual-branch architecture, our re-discovery of this crucial component is significant and important to guide future works on EPE. Please see the response to Q4 for further explanation.
>
> - **Designing Feature Map Tokens (FMT) for heatmap regression:** Ego-STAN leverages the spatio-temporal information for the conditional optimization model where the 3D pose relies on 2D heatmaps. Consequently, FMT is designed to aggregate spatial and temporal information from the time distributed CNN feature maps, and generalizes the concept of classification tokens for spatio-temporal data analysis [13]. As demonstrated by the significant improvements over the most difficult joints in Table 4 and Table 5 in Appendix A.1, this inductive bias for encoding spatio-temporal information is a novel approach for combating the severe distortions and occlusions in EPE.
>
> - **Extensive Ablations:** Our work undertakes a systematic analysis of each of these components – a first in the literature – via monte-carlo simulations. We release the code for the model, which is also the first for EPE – creating the first benchmark to spark further research in the area. Previous works have only released code pertaining to data processing [1,3,4]. As such, each of the variants in the ablation studies are extremely valuable for the area.
>
> - **Ego-STAN's Performance:** We achieve an **unprecedented 38.2% improvement** – that too in the joints that suffered from worst errors (hands and feet) in SOTA. This is important for critical downstream applications with direct impact on the fundamental progress of the area.
>
> > Q2. Although there is a graphical illustration of the FMT, it is still not clear about the specific operation. Concrete equations can be provided to make it clear about the actual operations.
>
> **Response:**
>
> Concrete equations and intuitive explanation of FMT can be found in sections 3.1 and 3.2 with specific intermediate dimensions. Equations 2, 3, 4, 8, and 9 all are related to FMT. We also added a section on intuitive explanation of FMT in Appendix A.3 based on reviewer's comments.
>
>
> >Q3. There are some details about the overall framework are missing. For example, how many (image) frames are used to reconstruct a 3D pose? Are the previous t frames used to reconstruct the (t+1) 3D pose? These settings should be made clear. Also, in the experiments, are these settings keep the same for all the competing methods?
>
> **Response:** In Appendix A.2.4 (under the "Data Augmentation" sub-heading), the reviewer will notice that our $T$ is selected as 5 with skip rate of 5. Additionally, Table 11 in the appendix illustrates our ablation studies with the sequence length and the skip rate. We have updated the manuscript to make this more obvious.

---

> > ### Public Comment · ~Anna_Clapp1 · 2023-02-16
> > **selfpose achieved similar results as yours just replacing resnet with unet**
> >
> > The papers you compared in table 1 are not SOTA. SelfPose achieved much better results.
> > @article{tome2020selfpose,
> >   title={Selfpose: 3d egocentric pose estimation from a headset mounted camera},
> >   author={Tome, Denis and Alldieck, Thiemo and Peluse, Patrick and Pons-Moll, Gerard and Agapito, Lourdes and Badino, Hernan and De la Torre, Fernando},
> >   journal={arXiv preprint arXiv:2011.01519},
> >   year={2020}
> > }

---

> > > ### Author Response · Authors · 2023-02-16
> > > **Response to Anna Clapp**
> > >
> > > Thanks a lot for your interest in our paper, we are absolutely thrilled to hear about your perspective. Self-Pose uses an encoder-decoder architecture showcasing how adding a third branch (rotation) and replacing the backbone from ResNet to UNet leads to improvements. Hence, in their ablations they isolate the effect of backbones. This is where our research questions are different. Specifically, we are not looking at role of different backbones, our goal is to build a model that can incorporate application-specific knowledge (to address self-occlusions and distortions) along with spatial and temporal information for 3D pose estimation. To this end, we propose a foundational approach to address the challenging task of ego-pose estimation which is backbone agnostic (we fix ResNet as a backbone for a fair comparison, but our method can work with any feature extractor such as UNET). Furthermore, we investigated the effect of encoder-decoder architectures, and found that it is in fact counter-productive.
> > > All of are results are reproducible (as opposed prior works). I hope this is helpful, please let me know if you want to discuss anything further.

---

> ### Author Response · Authors · 2022-11-19
> **Response to Reviewer kGU6 (Part 1 of 3)**
>
> We thank the reviewer for their questions. Below we comprehensively address each of the comments/clarifications in three parts (including this one). We hope these responses lead to a positive evaluation of our work, and we will be happy to clarify any additional follow-up questions/comments.

---

> ### Comment · Reviewer_kGU6 · 2022-12-04
> **Response to the author's response**
>
> I thank the authors for providing the responses to my comments and questions. However, I am still not convinced by the technical contribution/novelty and the experimental comparison. Unfortunately, I will keep my original rating.

---

> > ### Author Response · Authors · 2022-12-05
> > **Request for additional details about assessment**
> >
> > We thank the reviewer for their response. We comprehensively responded to all seven issues raised by the reviewer, exhaustively citing current work in the area, state-of-the-art, and highlighting how Ego-STAN surpasses each one of them by an unprecedented margin and fundamentally revolutionizes the entire area of ego-pose estimation.
> >
> > To understand and effectively help the reviewer in their assessment, we request that they elaborate on and identify a) which one of the responses did not convince them, and b) why. It is currently difficult to respond  and help with the process without knowing the specific references and aspects that the reviewer is basing their assessment on.
> >
> > Moreover, both Reviewers uHJW and cA4Q agree that the contributions such as a) effectiveness of the model, b) beating the baselines by a large margin, c) extensive abalations, d) detailed verification on three datasets, e) soundness and clarity in the paper, and f) reproducibility are significant to address this challenging task. Therefore, we look forward to receiving questions about specific aspects of the work to aid the reviewer's assessment in a meaningful way. We thank the reviewer again for their time.

---

### Official Review · Reviewer_uHJW · 2022-10-26

**Confidence:** 3
**Correctness:** 4
**Technical Novelty And Significance:** 3
**Empirical Novelty And Significance:** 3
**Recommendation:** 6

**Clarity, Quality, Novelty And Reproducibility:**

It would be good to release code or include a table to list framework structure in the appendix.

**Strength And Weaknesses:**

The entire paper is well-written and easy to follow. Their contributions are clear. They introduce simple but effective strategy, e.g. temporal spatial attention to aggregate information of image sequence, into the framework in the challenging egocentric human pose estimation in occlusions and strong distortions. The proposed Ego-STAN outpeforms several representative baselines largely.


**Summary Of The Paper:**

This work proposes a spatio-temporal transformer based model to estimate ego-centric 3D human pose in challenging scenarios, e.g. self-occlusions and strong distortion. The framework consists of feature map token (FMT), heatmap based representations and a direct 2D to 3D pose estimation module. They conducted experiments above two public datasets, and extensive ablation study to verify the effectiveness of each introduced module.

**Summary Of The Review:**

Their proposed framework can solve problems of pose estimation in challenging scenes, but the introduced modules have been already shown in previous works, so that I rate the novelty of this work as marginal above.

---

> ### Author Response · Authors · 2022-11-19
> **Response to Reviewer uHJW**
>
> Thank you for your encouragement, below we address the questions/comments raised by the reviewer. We hope these responses lead to a further positive evaluation of our work, and we will be happy to hear from you if you have any follow-up questions. Thank you for your time and feedback.
>
> >Q1. Clarity, Quality, Novelty And Reproducibility: It would be good to release code or include a table to list framework structure in the appendix.
>
> **Response:** We would like to gently point to the reviewer that **the code and implementation details to replicate and reproduce the results are presented in the supplementary material** (Section A.2.4 in Appendix). We also plan to host the code publicly on Github upon acceptance, which will also be, to the best of our knowledge, the first time where EPE models are made available to the research community; the state-of-the-art [1] only released the dataloaders to process the xR-EgoPose data. Additionally, the supplementary material also includes four representative videos – three for EPE and one for general pose estimation, further providing a qualitative demonstration of the superior EPE performance of Ego-STAN with extremely low jitter as compared to SOTA [1].
>
> > Q2. Summary Of The Review: Their proposed framework can solve problems of pose estimation in challenging scenes, but the introduced modules have been already shown in previous works, so that I rate the novelty of this work as marginal above.
>
> **Response:**
>
> **Novelty of the Spatio-temporal modeling:** The key contribution of this work is to leverage the spatio-temporal information to address the extreme self-occlusions and fish-eye distortions that plague EPE. As opposed to EPE, the outside-in undistorted view allows far less joint occlusions, making it easier for outside-in approaches to estimate occluded joint locations. Recent spatio-temporal outside-in works focus on regressing the 3D pose from a sequence of 2D keypoints (list of Euclidean joint locations) [5-8], with works also relying on additional joint visibility labels to address occlusions [9-11]. In contrast, the self-occlusions and distortions in our EPE make it extremely challenging to estimate 2D joint locations for 3D HPE, that to without access to visibility labels.
>
> Our work fundamentally addresses these unique challenges of EPE by estimating accurate 2D joint locations via heatmaps (which can model uncertainty in joint locations). Furthermore, Ego-STAN dynamically aggregating the temporal feature maps with FMT to counter the heavily self-occluded frames, accomplishing unprecedented performance gains. Additionally, nearly all other ego-pose methods are purely spatial-based [1,3,4], and therefore our spatio-temporal modeling is novel and a fundamental contribution to EPE.
>
>
> Further novel contributions of the paper include the following:
>
> - **Surprising result on Direct Regression:** Our contribution in this context is to re-introduce direct regression for pose estimation overlooked by recent works on EPE [1,2,9]. Specifically, we demonstrate via extensive ablations [Section 4.2, Table 3] that the dual-branch architecture proposed by SOTA [1] is complex and in fact hurts the pose estimation performance (a first in the literature to the best of our knowledge). Since recent methods [2,9] rely on SOTA’s dual-branch architecture, our re-discovery of this crucial component is significant and important to guide future works on EPE.
>
> - **Designing Feature Map Tokens (FMT) for heatmap regression:** Ego-STAN leverages the spatio-temporal information for the conditional optimization model where the 3D pose relies on 2D heatmaps. Consequently, FMT is designed to aggregate spatial and temporal information from the time distributed CNN feature maps, and generalizes the concept of classification tokens for spatio-temporal data analysis [13]. As demonstrated by the significant improvements over the most difficult joints in Table 4 and Table 5 in Appendix A.1, this inductive bias for encoding spatio-temporal information is a novel approach for combating the severe distortions and occlusions in EPE.

---

### Author Response · Authors · 2022-11-19
**General Comments to all Reviewers**

**Thank you for the feedback.** We would like to thank all the reviewers for their constructive feedback and suggestions. We address all comments separately in individual responses.

**Common abbreviations used in individual responses:**
- Feature Map Token (FMT)
- EgoPose Estimation (EPE)
- Human Pose Estimation (HPE)
- State-of-the-art (SOTA)

**Overview of the changes in the main script**.
We code the updates in blue in the updated manuscript and the appendices. Specifically, following is the list of main changes to the paper.

- Intuitive explanation of FMT in Appendix A.3.
- Difference between Ego-STAN's spatio-temporal modeling vs outside-in spatio-temporal models in Section 1
- Highlight the unique challenges of EPE and demonstrate why outside-in occlusion-robust methods are not relevant in Section 1, and Section 2.
- Reference for learnable positional embeddings in Appendix A.5
- Specifying the timestep being predicted in Sections 3.3 and 3.4.
- Report computational complexity in terms of MACs in Appendix A.4.



Additionally, we use the following references from the main script in the responses.

[1] Tome et. al. "xr-egopose: Egocentric 3d human pose from an hmd camera”. ICCV 2019

[2] Wang et. al. "Estimating egocentric 3d human pose in global space". ICCV 2021

[3] Xu et. al. “Mo 2 cap 2: Real-time mobile 3d motion capture with a cap-mounted fisheye camera”. VCG 2019

[4] Zhang et. al. “Automatic calibration of the fisheye camera for egocentric 3d human pose estimation from a single image”. WACV 2021

[5] Zheng et. al. “3d human pose estimation with spatial and temporal transformers”. ICCV 2021

[6] Zhang et. al. “Mixste: Seq2seq mixed spatio-temporal encoder for 3d human pose estimation in video”. CVPR 2022

[7] Cheng et. al. “3d human pose estimation using spatio-temporal networks with explicit occlusion training”. AAAI 2020

[8] Liu et. al. “Attention mechanism exploits temporal contexts: Real-time 3d human pose reconstruction”. CVPR 2020

[9] Cheng et. al. “Occlusion-aware networks for 3d human pose estimation in video”. ICCV 2019

[10] Jin et. al. “Occlusion-aware transformer for pose estimation in sparsely-labeled videos”. Arxiv 2022

[11] Liu et. al. “Explicit occlusion reasoning for multi-person 3d human pose estimation”. ECCV 2022

[12] Martinez et. al. “A simple yet effective baseline for 3d human pose estimation”. ICCV 2017

[13] Devlin et. al. “Bert: Pre-training of deep bidirectional transformers for language understanding” NAACL-HLT  2019.

[14] Dosovitskiy et. al. "An image is worth 16x16 words: Transformers for image recognition at scale". ICLR 2021

---

### Decision · Program_Chairs · 2023-01-20

**Decision:**

Reject

**Justification For Why Not Higher Score:**

The limited demonstration of the new module makes the submission marginally below the bar.

**Justification For Why Not Lower Score:**

N/A

**Metareview: Summary, Strengths And Weaknesses:**

This submission studies vision-based ego-centric 3D human pose estimation.  The authors proposed a spatial-temporal transformer with a feature map token (FMT) for better performance on this task. Reviewers appreciate the good performance and the presentation. However, there was a shared view that the advantages of the proposed module, in particular the FMT, are not clearly demonstrated and the performance gain is limited. The authors provided a detailed rebuttal, which helped to address the presentation issue. However, reviewers remain not convinced about the value of the technical innovation FMT. The AC read the reviews, rebuttals, and the AC note carefully, and looked at the paper in detail. Finally, the AC recommends rejection as the limited demonstration of the new module makes the submission marginally below the bar. The authors are encouraged to further improve the paper for the next venue.